# VISUALIZING HIGH-DIMENSIONAL TRAJECTORIES ON THE LOSS-LANDSCAPE OF ANNS

## ABSTRACT

Training artificial neural networks requires the optimization of highly non-convex loss functions. Throughout the years, the scientific community has developed an extensive set of tools and architectures that render this optimization task tractable and a general intuition has been developed for choosing hyper parameters that help the models reach minima that generalize well to unseen data. However, for the most part, the difference in trainability in between architectures, tasks and even the gap in network generalization abilities still remain unexplained. Visualization tools have played a key role in uncovering key geometric characteristics of the loss-landscape of ANNs and how they impact trainability and generalization capabilities. However, most visualizations methods proposed so far have been relatively limited in their capabilities since they are of linear nature and only capture features in a limited number of dimensions. We propose the use of the modern dimensionality reduction method PHATE which represents the SOTA in terms of capturing both global and local structures of high-dimensional data. We apply this method to visualize the loss landscape during and after training. Our visualizations reveal differences in training trajectories and generalization capabilities when used to make comparisons between optimization methods, initializations, architectures, and datasets. Given this success we anticipate this method to be used in making informed choices about these aspects of neural networks.

## 1 INTRODUCTION

Artificial neural networks (ANNs) have been successfully used to solve a number of complex tasks in a diverse array of domains, such as object recognition, machine translation, image generation, 3D protein structure prediction and many more. Despite being highly overparameterized for the tasks they solve, and having the capacity to memorize the entire training data, ANNs tend to generalize to unseen data. This is a spectacular feat since the highly non-convex optimization typically encountered in them should (theoretically) be a significant obstacle to using these models (Blum & Rivest, 1993). Questions such as why ANNs favor generalization over memorization and why they find good minima even with intricate loss functions still remain largely unanswered. One promising research direction for answering them is to look at the loss-landscape of deep learning models. Recent work tried to approach this task by proposing various visualization methods. An emerging challenge here is how to look at such an extremely high dimensional optimization landscape (linear in the number of parameters of the network) with respect to minimized loss.

In past work, loss functions and their level lines were visualized via random directions starting at a minimum, or by means of linear methods like PCA. In some case, this approach proved effective in uncovering underlying structures in the loss-landscape and link them to network characteristics, such as generalization capabilities or structural features (Keskar et al., 2016; Li et al., 2018). However, these methods have two major key drawbacks: **(1)** they are linear in that they only choose directions that are linear combinations of parameter axes while the loss landscape itself is highly nonlinear, and **(2)** they choose only two among thousands (if not millions) of axes to visualize and ignore all others.

In this work, we utilize and adapt the PHATE dimensionality reduction method (Moon et al., 2019), which relies on diffusion-based manifold learning, to study ANN loss landscapes by visualizing the evolution of network weights during training in low dimensions. In general, visualizations like PHATE Moon et al. (2019) are specifically designed to squeeze as much variability as possible into

two dimensions, and thus provide an advantage over previous approaches. In particular our choice of using PHATE over other popular methods, such as tSNE (van der Maaten & Hinton, 2008), is due to its ability to capture global and local structures of data, and in particular to keep intact the training trajectories that are traversed through during gradient descent. Indeed, during training, the high-dimensional neural networks weights change significantly while remaining on a connected manifold defined by the support of viable configurations (e.g., with sufficiently low training loss), which we refer to when discussing the geometry of the loss landscape. We show that PHATE is suitable to track such continuous weight trajectories, as opposed to tSNE or UMAP that tend to shatter them. Moreover, our approach provides general view of relevant geometric patterns that emerge in the high-dimensional parameter space, providing insights regarding the properties of ANN training and reflecting on their impact on the loss landscape.

**Contributions:** We propose a novel loss-landscape visualization based on a variation of PHATE, implemented with cosine distance in Section 4. Our method is, to our knowledge, different from all other proposed methods for loss visualization in that it is naturally nonlinear and captures data characteristics from all dimensions. In Section 5.1, we show that our method uncovers key geometric patterns characterizing loss-landscape regions surrounding good and bad generalization optima, as well as memorization optima. Finally, we establish the robustness of our method by applying it to numerous tasks, architectures, and optimizers in Sections 5.3 and 5.2, suggesting that our method can be used in a consistent manner to validate training and design choices.

## 2 RELATED WORK

Loss landscape visualization methods have been proposed in numerous contexts. Goodfellow et al. (2014) proposed the "linear path experiment" where the loss of an ANN is evaluated at a series of points $\theta = (1 - \alpha)\theta_i + \alpha\theta_f$ for different values of $\alpha \in [0, 1]$ and $\theta_i, \theta_f$ corresponding to the initial parameters of the model and the found optima in parameter space respectively. This one-dimensional linear interpolation method has allowed them to show that popular state of the art ANNs typically do not encounter significant obstacles along a straight path from initialization to convergent solution. They also used the method to visualize the loss along directions connecting two distinct minima and to show that these are linearly separated by a region of higher valued loss.

This method was further developed by Im et al. (2016), who adapted it to enable the visualization of two-dimensional projections of the loss-landscape using barycentric and bilinear interpolation for groups of three or four points in parameter space. This analysis method has allowed them to establish that despite starting with the same parameter initialization, different optimization algorithms find different minima. Furthermore, they noticed that the loss-landscape around minima have characteristic shapes that are optimizer-specific and that batch-normalizations smooths the loss function.

More recently, Li et al. (2018) have addressed the *scale invariance* and *network symmetries* problems discussed in Neyshabur et al. (2017); Dinh et al. (2017), which prevented meaningful comparisons between loss-landscape plots from different networks. They proposed 1D and 2D linear interpolation plots, similar to past techniques, but where they used filter-wise normalized directions to remove the scaling effect. This method has allowed them to visualize and compare the regions on the loss-landscape surrounding minima coming from multiple networks in a meaningful way and to correlate the "flatness" of the region to the generalization capabilities of the corresponding network. Furthermore, they studied the effects of the network depth, width and the presence of skip connections on the geometry of the loss-landscape and on network generalization.

The importance of loss landscape visualization methods like the one presented in this paper increases with the growing scientific community interest in furthering our understanding of ANNs. As our understanding of this landscape gets deeper, we start uncovering more and more high-dimensional and complex geometric and topological characteristics. For instance, Draxler et al. (2018) have found nonlinear pathways in parameter space connecting distinct minima, along which the training and test errors remain small and losses are consistently low. This suggests that minima are not situated in isolated valleys but rather on connected manifolds representing low loss regions. However, such characteristics are intrinsically high-dimensional, making linear methods inadequate to visualize these structures. Even in standard applications of the linear methods one inevitably asks if the thousands (or

even millions) of unseen directions do not hide critical features of the landscape, and if the visualized linear path is relevant to what happened during training.

In this paper, we suggest the use of modern dimensionality reduction techniques to study the complex loss-landscape of ANNs. Such techniques have been extensively used in recent years to successfully study the internal learned representations of deep networks and highlight their complex geometric structures and intrinsic dimensionality (Gigante et al., 2019; Horoi et al., 2020; Recanatesi et al., 2018; Farrell et al., 2019; Maheswaranathan et al., 2019), and here we further advance this line of work to provide new applications and insights in the study of ANN optimization and generalization.

## 3 PRELIMINARIES: PHATE DIMENSIONALITY REDUCTION & VISUALIZATION

Given a $n \times m$ data matrix $\mathbf{N}$, where $n$ is the number of data points and $m$ the number of features, PHATE computes low-dimensional embeddings of the data points (dimension $d$ to be specified by the user) which preserve both their global and their local structure. It first computes the pairwise distance matrix $\mathbf{D}$ using a specified distance function $\phi$, each element of the matrix being given by $\mathbf{D}_{ij} = \phi(\mathbf{N}_{i,:}, \mathbf{N}_{j,:})$. To better capture local affinities between data points, an affinity matrix $\mathbf{A}$ is computed using an $\alpha$-decaying kernel with a locally-adaptive bandwidth $\epsilon_{k,i}$ corresponding to the $k$-NN distance of the $i$-th data point. The elements of $\mathbf{A}$ are given by:

$$\mathbf{A}_{i,j} = K_{k,\alpha}(i,j) = \frac{1}{2} \exp\left(-\left(\frac{D_{i,j}}{\epsilon_{k,i}}\right)^{\alpha}\right) + \frac{1}{2} \exp\left(-\left(\frac{D_{i,j}}{\epsilon_{k,j}}\right)^{\alpha}\right)$$

The decaying factor $\alpha$ regulates the decay rate of the kernel (smaller $\alpha \Rightarrow$ kernel with lighter tails), $\alpha = 2$ corresponding to the Gaussian.

The affinity matrix is then row-normalized to obtain the diffusion operator $\mathbf{P}$, a row-stochastic Markov transition matrix with element $\mathbf{P}_{i,j}$ giving the probability of jumping from the $i$-th to the $j$-th data point in one time step. One of the reasons PHATE excels at capturing global structures in data, especially high-dimensional trajectories and branches, is that it leverages the diffusion operator (also used, Coifman & Lafon, 2006, to construct diffusion maps) by running the implicit Markov chain forward in time. This is accomplished by raising the matrix $\mathbf{P}$ to the power $t$, effectively taking $t$ random walk steps and revealing intrinsic geometric structure through the affinity of the data at a scale that grows with the number of steps. The optimal value for $t$ is automatically chosen to be the knee point of the von Neumann entropy of $\mathbf{P}$.

To enable dimensionality reduction while retaining diffusion geometry information from the operator, PHATE leverages information geometry to define a pairwise *potential distance* as an M-divergence $\mathbf{ID}_{i,j} = \| \log P_{i,:} - \log P_{j,:} \|_2$ between corresponding $t$-step diffusion probability distributions of the two points, which provides a *global context* to each data point. The resulting information distance matrix $\mathbf{ID}$ is finally embedded into a visualizable low dimensional (2D or 3D) space by metric multidimensional scaling (MDS), thereby squeezing the intrinsic geometric information into two dimensions (as is also done in tSNE), rather than producing a high dimensional alternative coordinate system like PCA or diffusion maps.

We reason that since PHATE learns a data manifold via diffusion, and places points in the global context of other points, it can keep trajectories connected from beginning to end without shattering them. Indeed, this property was established quantitatively in Moon et al. (2019) via a Denoised Manifold Affinity Preservation (DeMAP) metric. Here, it leads to the preservation trajectory flow directions and relations between them, as demonstrated in Figure 1, in comparison with other methods, on trajectory-like data and ANN training trajectories, as well as in our results presented in the remainder of the paper.

## 4 METHODS

The loss-landscape is a high dimensional function $f(\theta)$ that assigns a loss value to every possible vector $\theta$ in the high-dimensional parameter space of an ANN denoted $\Theta$. The dimensionality of the space renders the task of completely visualizing the loss-landscape virtually impossible. By gathering locally connected loss-landscape patches throughout training and around minima found via gradient descent, we hope to reconstruct manifolds on which these trajectories lie. Despite

the thousands, if not millions, of dimensions of the parameter space $\Theta$, we hypothesize that these trajectories and manifolds are intrinsically of low dimensional and thus PHATE, which reduced dimensionality via manifold learning and information geometry, will help facilitate their visualization in two specific settings: 1. To characterize the region of the loss-landscape surrounding minima; and 2. To simultaneously visualize multiple training trajectories corresponding to different parameter initializations and optimizers in order to make training choices.

In setting **(1)** we use the following "jump and retrain" experiment. Given a trained ANN and a minimum found during training with an optimizer:

- Let $\theta_o$ represent the vector of network parameters at the minimum;
- For $seed \in \{0, 1, 2, 3, 4\}$:
    - For $step\_size \in \{0.25, 0.5, 0.75, 1\}$:
        1. Choose the random vector $v_{seed}$ in the parameter space of the network $\Theta$ corresponding to the random seed $seed$;
        2. Filter normalize the vector as proposed in Li et al. (2018) to obtain $\overline{v}_{seed}$;
        3. Set the network parameters at each jump initialization to be $\theta_{\text{jump-init}} = \theta_o + step\_size \cdot \overline{v}_{seed}$;
        4. Retrain for 50 epochs using optimizer with 0.9 momentum on the original training set;
        5. Record network parameters $\theta$ for each seed and step size combination and at the end of each epoch.
- Apply PHATE to the data matrix $[\theta_{seed,step\_size,epoch}]$ and visualize the embeddings.

In setting **(2)**, we simultaneously visualize multiple training trajectories corresponding to different parameter initializations and optimizers. Here, we simply follow the optimizer from initialization to convergence, and record the parameter vector at each step.

We note that methods that pick random directions or surfaces around the minima projected to random directions are not able to visualize entire trajectories as the spaces and planes change. This is a distinct advantage of using all of the dimensions of the parameter space. In both settings, PHATE has allowed us to bypass the key drawbacks of previously proposed linear interpolation methods by: 1. Capturing variance in data from all relevant dimensions and embed it in a low-dimensional space; and 2. preserving high-dimensional trajectories and global structures of data in parameter space. All modern dimensionality-reduction techniques would have achieved **(1)** in one way or another and with various degrees of success. The reason our method is based on cosine-distance PHATE, is to better accomplishes **(2)** by preserving trajectories and global relationships in high-dimensional data to a greater extent than any other state-of-the-art dimensionality-reduction technique. Figure 1 demonstrates this by showing a comparison of multiple such techniques, namely PHATE, PCA, t-SNE (van der Maaten & Hinton, 2008) and UMAP (McInnes & Healy, 2018), and how they each embed the results of a jump and retrain experiment (Figure 1**A** and **B**) and an artificial data set having a tree-like structure (Figure 1**C**) in a 2D space.

While some trajectory-like structure is visible in all low-dimensional embeddings, only PHATE properly captures intra-trajectory variance. PHATE is also the only technique that captures the global relationships in between trajectories while t-SNE and UMAP have a tendency to cluster points that are close in parameter space and disregard the global structure of the data. On the artificial data set, what we observe is that the embeddings of the linear method PCA are highly affected by the noise in the data while t-SNE and UMAP have a tendency of shattering trajectories that should be connected.

## 5 Case studies and applications

### 5.1 Generalization vs. Memorization

Making sure that machine learning models can generalize to new data is one of the most important steps in solving any data analysis task, and is a core problem in the field. It is so important that oftentimes large fractions of the scarcely available data are not used for training but rather for testing, to ensure that models indeed learn to solve the task in a relevant manner without overfitting their training data. Arguably one of the most important and general questions in machine learning at the moment is: by looking only at a given trained model and its training data, can we predict how well the model will perform on unseen data? There is even an entire NeurIPS 2020 competition entirely focused on answering this question (Jiang et al., 2020).

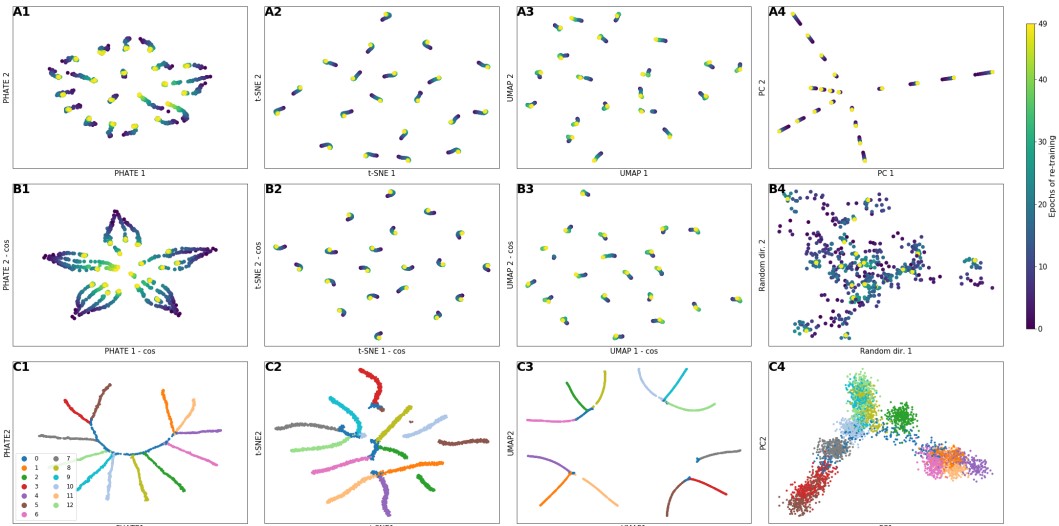

Figure 1: 2D embeddings of the jump and retrain experiment results for a WResNet28-2 network using the modern dimensionality reduction techniques PHATE (**1**), t-SNE (**2**), UMAP (**3**) using euclidean (**A**) and cosine (**B**) distances and the linear methods PCA (**A4**) and random plane projection (**B4**). **C:** embeddings found using the same techniques of an artificial data set having a fully connected tree-like structure. We see that PHATE consistently retains continuous trajectory structures while other embeddings (tSNE/UMAP) shatter the structure, or show chaotic trends because of uninformative projections to low dimensions (PCA, random directions). See Appendix A for higher resolution.

Previous work offers a partial and vague answer to this question by linking the "flatness" of neural network optima to the extent to which the network is able to generalize. This idea was initially presented in Hochreiter & Schmidhuber (1997) and has recently been the focus of numerous papers (e.g., Chaudhari et al., 2016; Keskar et al., 2016; Swirszcz et al., 2016; Dinh et al., 2017; Li et al., 2018). Flat minima in parameter space, around which the value of the loss function is fairly constant, are believed to correspond to configurations where the network generalizes well to the unseen data. Sharp minima, on the other hand, correspond to ones where the network is expected to generalize poorly. Following the underlying idea of these propositions, we believe that studying the geometry of the loss landscape around minima has the potential of uncovering key characteristics about the network's ability to generalize. Here, we show that jump-and-retrain experiments combined with our visualization method reveal stark differences between neural network configurations that memorize (or overfit) versus ones that generalize.

For our empirical results we trained multiple WResNet28-2 networks (wide ResNet of depth 28 and 2 times as wide as a standard ResNet (He et al., 2016)) presented in Zagoruyko & Komodakis (2016) on the CIFAR10 dataset (Krizhevsky et al., 2009) using batches of size 128 for 200 epochs with the SGD optimizer with 0.9 momentum and a learning rate of 0.1 which decays by a factor of 10 at epoch 150. In some cases we did not use data augmentation and weight decay and in others we used 1e-4 weight decay and random crops and horizontal flips for the training data. Adding the data augmentation and the weight decay has allowed us to find optima that generalize better, increasing the accuracy on the test data from around 85% to around 95%. We will be referring the optima with ~95% test accuracy as "good optima" since they generalize better, and that with lower test accuracy as "bad optima". Using the same training procedure as the one for the bad optima, we also trained a WResNet28-2 network for memorization by completely randomizing the labels of the training data. This modification was used in Zhang et al. (2017); Arpit et al. (2017); Krueger et al. (2017) to show that neural networks have the ability to completely memorize the training data sets. We then ran the jump-and-retrain experiment for these three types of minima and visualized the results using PHATE in Figure 2. This allows us to analyze and compare the regions in parameter space around the good, bad and memorization minima, which all achieve ~0 loss on their respective training sets.

Despite only a ~7% difference in their accuracy on the test data set, the PHATE embedding of the trajectories surrounding each generalization minimum is significantly different as seen in Fig. 2d. The minimum that shows good generalization has a distinctive star or flower shaped pattern. This

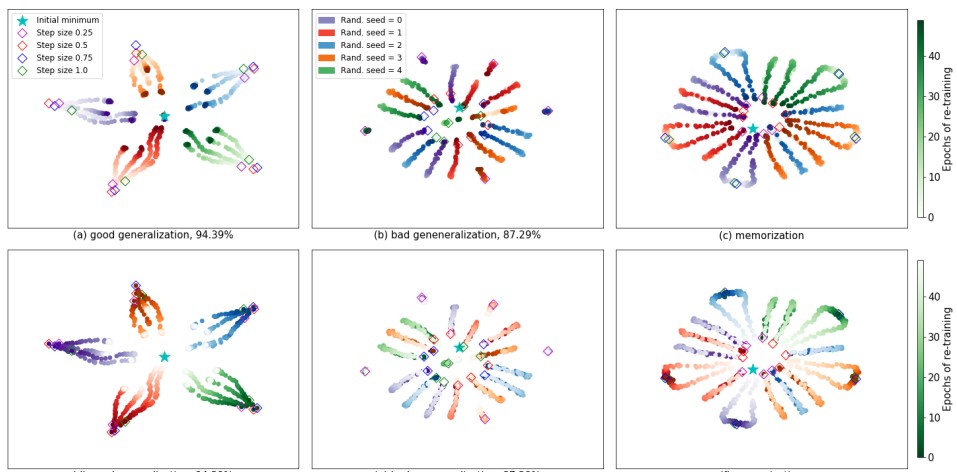

Figure 2: 2D PHATE embeddings of the jump and retrain experiment conducted on one good (**a,d**) and one bad (**b,e**) generalization minima that achieve 94.39% and 87.29% accuracy on the test data respectively, as well as one memorization minimum (**c,f**). The initial points of the jumps (before retraining) are marked by diamonds of different colors based on $step\_size$ in all plots. All trajectories are colored by their respective direction of the initial jump. Plots **a,b,c** have ascending hue, meaning the color gets darker as retraining progresses. Plots **d,e,f** have descending hue, meaning the color gets whiter as retraining progresses. In **d** and **e** the points are colored by the direction ($seed$) of the jumps in parameter space and the color grows darker as retraining progresses. In contrast to more continuous trajectories returning to near the optimum in the good generalization case, memorization displays a more random pattern where weights move out before moving back, often switching direction during retraining. Bad generalization appears to be more similar to memorization, where no continuous trajectories were observed. See Appendix B for a higher resolution version.

pattern in a jump-and-retrain experiment indicates that even when points are thrown far from the minimum, the landscape is wide enough that they return to it. Moreover, they tend to return in the same direction they were pushed out, leading to a flower-petal convergence pattern. Such a structure is the high-dimensional equivalent to a "wide valley" described in past work studying flatness of optima. While the starting points are all set on five distinct directions from the minimum, the plotted retraining trajectories are not restricted to any subspace of the parameter space.

In the bad minimum on the other hand, the trajectories start off near the middle of the plot (darker points in the middle of Fig. 2b) but during the retraining, the trajectories diverge toward the edges before coming back towards the middle of the plot. This outward movement is stark contrast with the consistent retraining trajectories surrounding the good generalization minimum, which immediately return to the valley. This indicates that the region around the bad-generalization minimum is most probably not a wide (or flat) high-dimensional valley and contains a nonnegligible amount of nonconvexities, preventing training trajectories to systematically converge towards the minimum. The "memorization" minimum plot Fig. 2c looks similar to the bad generalization plot, with trajectories that go outward at small step sizes of the "jump." However, curiously at larger step sizes of the "jump," the trajectories seem to return without going outward first, but they do not return immediately, i.e., they show some lateral movement, potentially indicating bumpiness in the landscape that they are avoiding. Thus we see a clear pattern in the good generalization minima of inward movement upon the jump and retrain, and more outward behavior in the bad generalization and memorizations. See supplemental video for an animation of these trajectories, and Appendix B for trajectories colored by loss.

## 5.2 OPTIMIZATION

We analyzed three stochastic gradient-descent optimization methods commonly used for training ANNs: SGD (vanilla), SGD with momentum (SGD_M), and Adam. The major difference is that Adam uses adaptive step sizes based off previous iterations while SGD and SGD_M use fixed step

sizes. We are interested in answering questions including how the network evolves during training with different optimizer and whether different optima are reached at the end of training.

We trained Wide-ResNet networks of depth 28 and twice as wide, which is the same as in the previous section. The same initial learning rate of 0.1 and learning rate decay schedule is employed with each optimizer, data augmentation is used to assist training while no weight decay was used. All experiments achieve close to 100% training accuracy and > 90% test accuracy. A fixed set of five seeds is used across optimizers.

First we examine individual trajectories of network weights from the same seed using different optimizers (Fig. 3). Compared to the smooth trajectory from using Adam, SGD with or without momentum seems to exhibit scattered patterns. It should be noted that the magnitudes of network weights learned by Adam are much larger than by SGD or SGD_M (Appendix D). The evolution and resulting network weights from the Adam optimizer appears to differ more significantly from SGD or SGD_M, where the resulting parameters appear to be further away from the initialization.

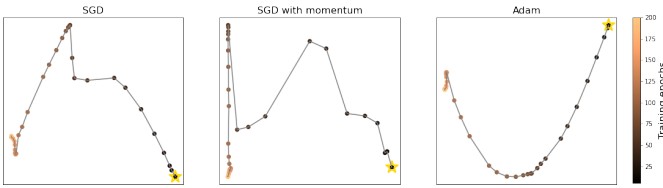

Figure 3: Comparing PHATE embedding of network weights during training using different optimizers. The same seed was used across optimizers, Adam produces a more smooth trajectory.

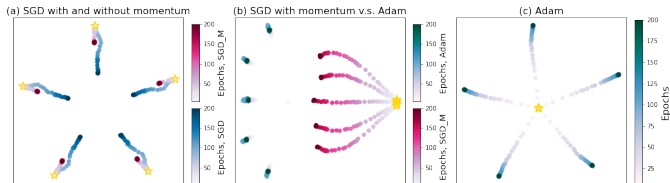

Figure 4: PHATE embedding (cosine distance) of network parameters during training using different optimizers (SGD, SGD_M, and Adam). The five initialization are marked by golden stars. (a),(b)Different optimization algorithms reach distinct minima that all lie in a disc-shaped region. ADAM seems to search an orthogonal space of the loss landscape from SGD with momentum, as seen by the "disappearance" of the trajectory, (c) The Adam trajectories, plotted by themselves are continuous, however distances between final weights reached by Adam are much larger than other optimizers.See Appendix E for higher resolution images

In Fig. 4 the distinctiveness between optima reached from different initializations and optimizers are clearly shown. We see in the Fig. 4(a) that when starting from the same seed, SGD and SGD with momentum travel in similar directions. However, we see a shorter and straighter training path with momentum. Without momentum, a similar minima is reached. However, the path towards the minima is longer with more moves sideways, indicating that the space is searched less effectively. In Fig. 4(b) the fact that Adam travels so far in the weight space, causes the initial points look very close together. From the initial points, the SGD with momentum travels in shorter paths towards different minima. However, Adam seems to exhibit a disconnected jump, potentially flying off into an orthogonal space to what is visualized, but interestingly returning to the same region of minima as SGD_M. The seemingly discontinuity of the trajectories in the middle plots indicates that Adam searches space very differently than SGD. Fig. 4(c) shows that the Adam trajectories are indeed connected in a continuous line, so the trajectory itself is not discontinuous, but rather in a different subspace than the SGD_M trajectory.

## 5.3 ROBUSTNESS

One of the main problems encountered in previous attempts to visualize ANN loss landscapes resides in the difficulty of comparing them across tasks and initializations. Here, we show that our

visualization approach, applied to the results of jump-and-retrain experiments, alleviates this concern. To this end, we trained multiple WResNet28-2 networks by following the procedure described in Sec. 2, and starting from multiple parameter initializations. For each initialization, we found one good (~94% test acc.) and one bad optima (~86% test acc.), giving the jump-and-retrain results shown in Fig. 5. Further, we also trained a similar network on the CIFAR100 dataset, following the same training procedure, except here the data augmentation was comprised not only of random horizontal flips and crops, but also of random rotations of up to 45° in each direction and random modifications to the brightness, contrast, saturation and hue of the images. Since CIFAR100 poses a significantly harder classification task than CIFAR10, our networks only reaches ~60% test accuracy for the bad optimum trained without data augmentation and weight decay, and ~70% test accuracy for the good optimum. The fact that the networks only reached 70% accuracy is just a reflection of the complexity of the task, with better accuracy requiring a better fine-tuning of the architecture and the training procedure. The results of these experiments are shown in Fig. 5

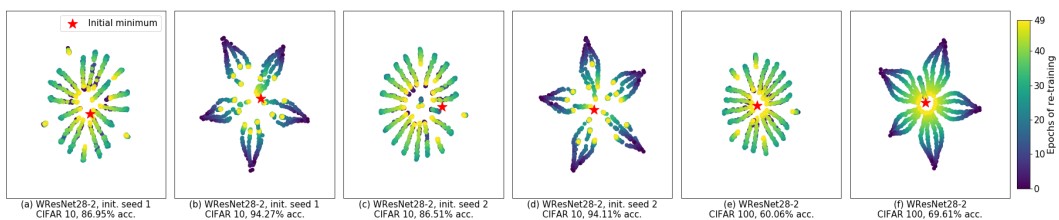

Figure 5: 2D PHATE embeddings of the jump and retrain experiment ran on different initialization for the WResNet28-2 network on CIFAR10 (**a,b,c and d**) and on CIFAR 100 (**e and f**), colored by epoch of retraining. Patterns that resemble bad generalization are exhibited in the first plot for all three settings as in (a), (c) and (e), while good generalization patterns are shown in the second plots (b), (d) and (f). It demonstrates that our approach is robust across tasks. See Appendix C for a higher resolution version.

The plots for different initializations of the WResNet28-2 are consistently showing the same star-shaped patterns, characterizing the retraining trajectories surrounding minima that generalize well. The plots for the bad minima also share a similar pattern, not only across initialization but also across tasks. Strikingly, the star-shaped pattern is even better defined in Fig. 5f, corresponding to the good optimum of WResNet28-2 trained on CIFAR100. This offers a key insight into the effect of regularization on the geometric characteristics of the found minima. It seems that regularization techniques enhance generalization by forcing the optimizer into wider valleys.

## 6 DISCUSSION AND CONCLUSION

We proposed a novel approach to visualize ANN loss landscapes based on the state-of-the-art PHATE dimensionality reduction, which is able to capture branch-like structures in high-dimensional data in two dimensional representations. Our approach enables geometric exploration of retrained trajectories surrounding generalization and memorization optima, found via ANN training, to provide insight into generalization capabilities of the network. Further, it enables meaningful comparison of training trajectories across optimizers. Finally, our proposed paradigm is demonstrably consistent and robust across initializations and supervised learning tasks. We expect in future work this visualization approach to enable more methodical paradigms for the development of deep models that generalize better, train faster, and provide fundamental understanding of their capabilities.

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

APPENDIX

A

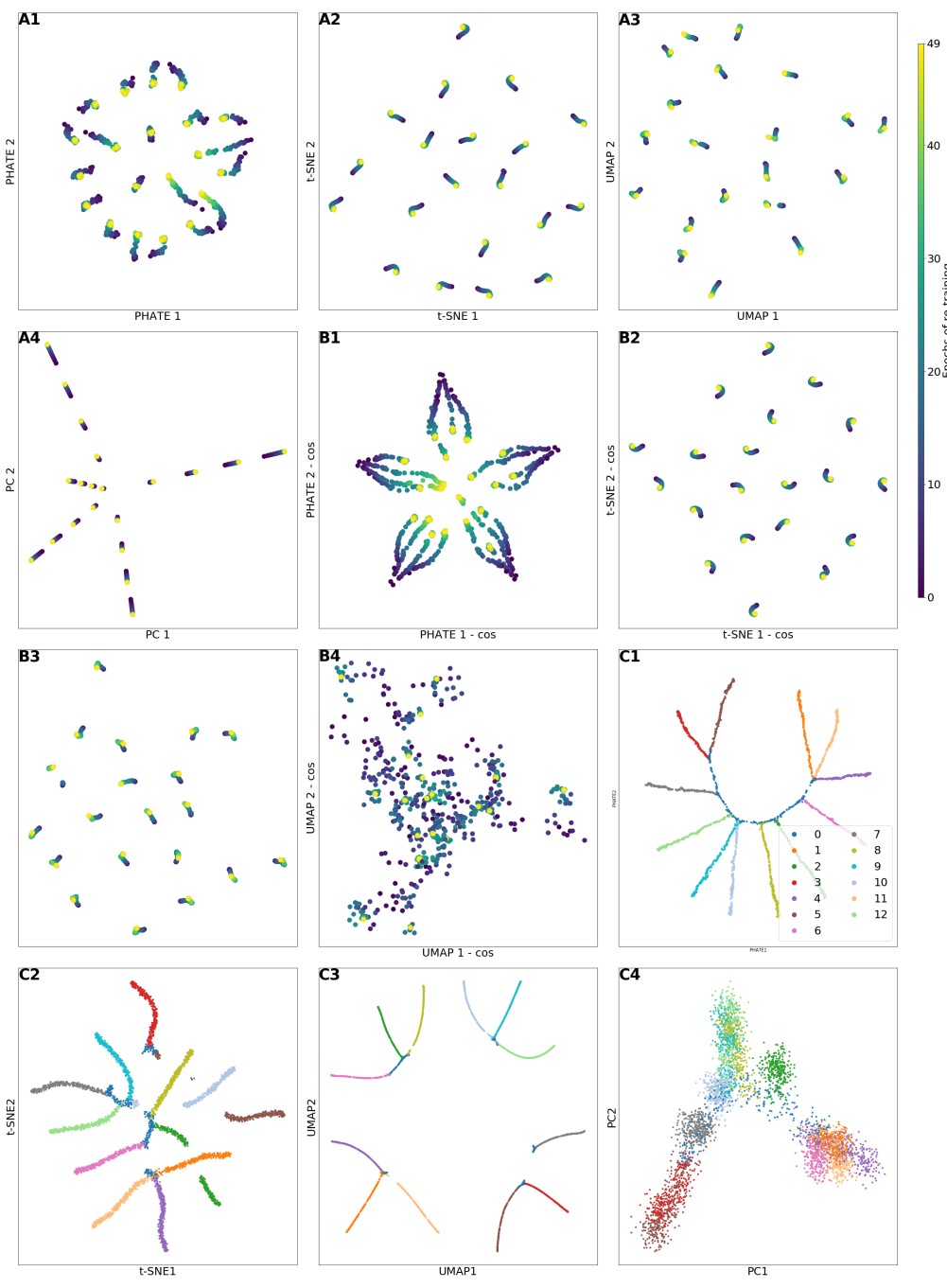

Figure 6: Higher resolutionversion of Figure 1

B

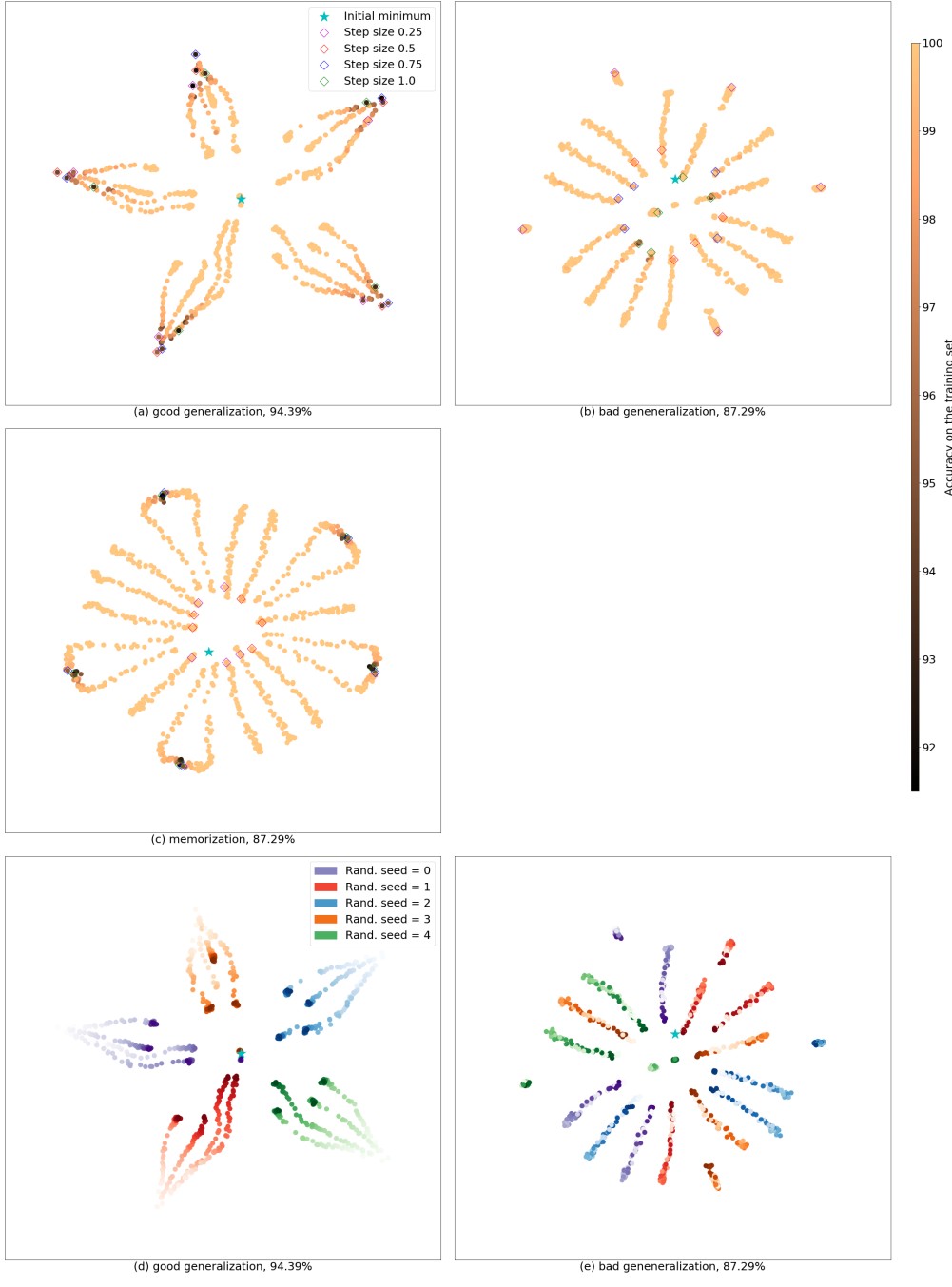

Figure 7: Higher resolution version of Figure 2

C

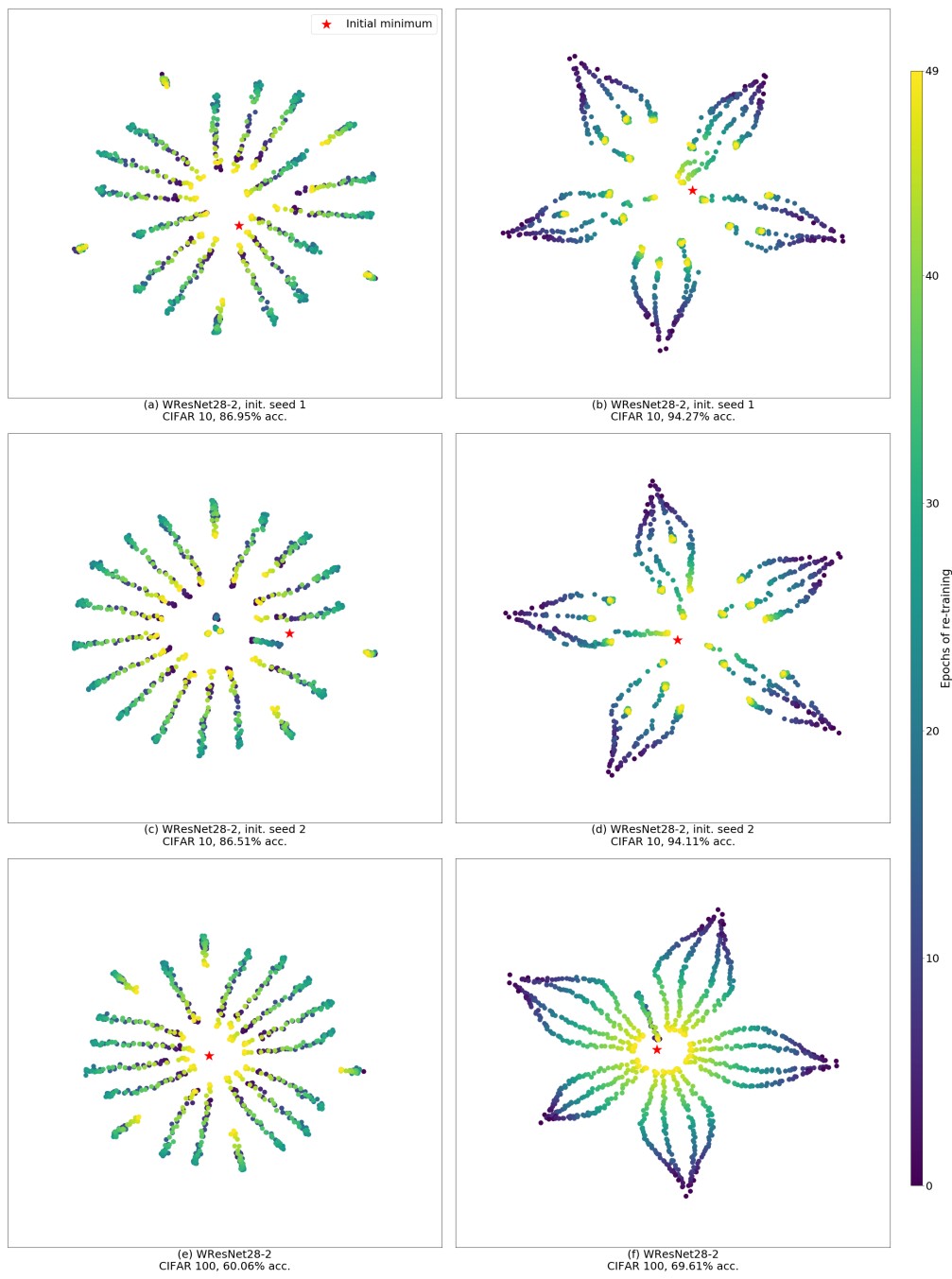

Figure 8: Higher resolution version of Figure 5

D

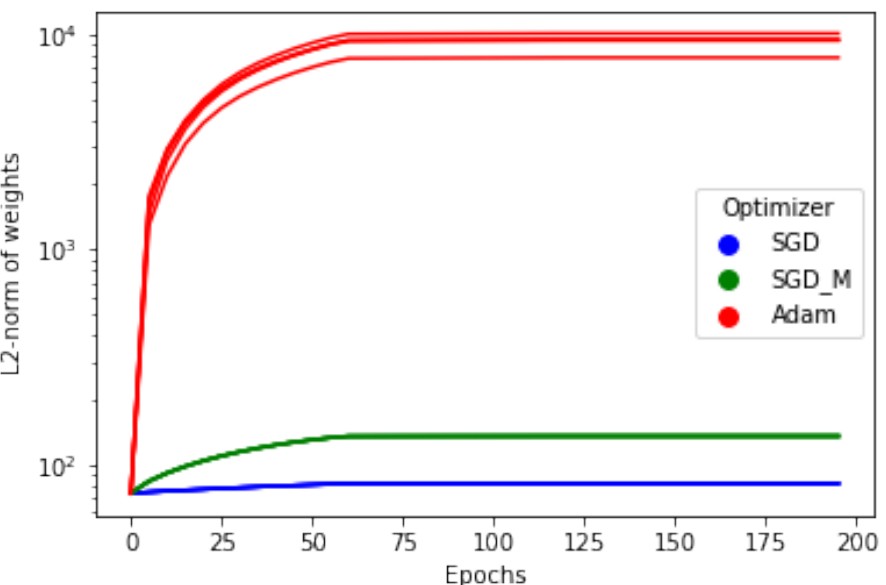

Figure 9: L2 norm of network weights during training using different optimizers.

E

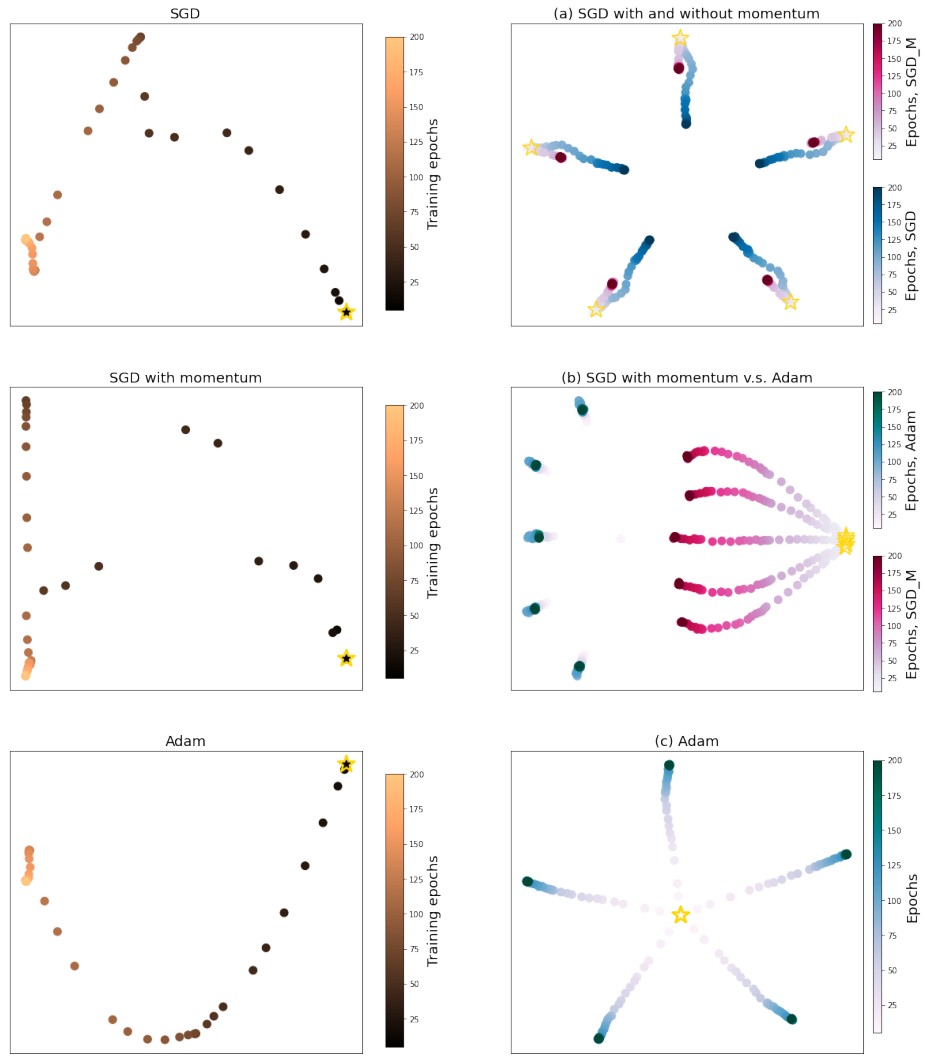

Figure 10: Higer resolution of Figure 3and Figure 4

