# OpenReview forum: "Visualizing High-Dimensional Trajectories on the Loss-Landscape of ANNs"
_ICLR.cc/2021/Conference — Reject_

### Official Review · AnonReviewer3 · 2020-10-18
**Presents an improved approach to visualizing multiple learning trajectories, which is useful in understanding NN generalization. Approach is correct and useful, but resulting insights are a little underwhelming.**

**Rating:** 6
**Confidence:** 5

**Review:**

### Paper Summary

This paper uses PHATE to visualize the progression of neural net parameters during learning in neural networks to provide insight into generalizable vs. non-generalizable minima, and the behaviors of different optimization algorithms.  PHATE is an improvement over previous visualization techniques due to its approach to finding a manifold, allowing it to plot in two dimensions multiple trajectories that do not otherwise share a plane.  This is then used to plot trajectories in "jump and retrain" experiments, in which a minimum is found, and then perturbations made to the network parameters, before restarting training.  It is shown that in the networks experimented on, minima with good test set performance reliably funnel the new learning trajectories into the same minimum, while minima with poor test set performance see the perturbed initializations find other minima, thus demonstrating the "flatness vs sharpness" of minima.  Trajectories produced by SGD, SGD with momentum, and Adam are also compared.  Adam is shown to travel further, but along a smoother trajectory.

### Originality

The visualization approach is not new, but is new to neural network trajectories.  The visualizations effectively confirm suspected properties of neural networks, but do not offer anything particularly new.  The behaviors of learning algorithms are demonstrated, but not much discussed, so little new is learned.

### Significance

Visualization is important to understanding the behaviors of high-dimensional learning trajectories, and this is better for plotting multiple trajectories than previous approaches.  It is an effective approach, and is superior at demonstrating known neural network behaviors than others.  Insights are not significant, but the visualization of them is improved.

### Clarity

Very clearly written, and easy to understand.

### Quality

Enjoyable paper, which offers a new, effective tool, but no real new insights into neural networks.  The jump-and-retrain experiments are effective and well-chosen.  The demonstrations of learning algorithm are not as effective, and I'm not sure what to take away from them.  I invite the authors to help me find the significance of these experiments.

---

> ### Author Response · Authors · 2020-11-25
> **Author response to Reviewer 3**
>
> We thank the reviewer for his/her appreciation of the effectiveness and importance of the proposed visualization method as well as for the positive comments regarding the clarity of the text and overall appreciation of our paper. We understand the need for stronger support elements to showcase the utility of our methods.
>
> As for the significance of the experiments on learning algorithms, the goal was to qualitatively compare the trajectories of different optimizers. Momentum SGD and Adam go further away from the initializations and travel a longer distance than standard SGD (as shown by the norm in Appendix D). However, the path travelled by Adam or SGD_M may contain a lot less variability (as expected momentum has a "normalizing" or regularizing effect on the trajectory) while the noise inherent in SGD makes the path less continuous. Our visualization method picks up on this variability as shown in Figure 4.

---

### Official Review · AnonReviewer4 · 2020-10-27
**Using PHATE to visualize gradient descent trajectories**

**Rating:** 4
**Confidence:** 4

**Review:**

The paper suggests using PHATE, a modern dimensionality reduction method, for visualizing the training trajectories of deep networks. It argues that PHATE visualizations can bring to light interesting aspects of the training dynamics that are missed by other dimensionality reduction algorithms because PHATE does a better job at preserving both local and global structure in the data. This is not the first application of PHATE in the context of deep learning, but to my knowledge it is the first application to deep learning trajectories.

The paper shows that PHATE visualizations of train-and-jump trajectories (where the model is repeatedly pushed away from a minimum and then allowed to retrain) can have features that are correlated with how well a model generalizes. It also shows that different optimization algorithms can lead to distinguishable visual features.

While the use of better visualization techniques may lead to better understanding of neural network dynamics, I am not convinced that the paper succeeds in making this case:

1. The generalization results (Figure 2) are interpreted using known results about the generalization properties of flat vs. sharp minima. These results are nice, and are potentially useful. However, they were only demonstrated in a few settings. Do these results hold more generally, when using other architectures and different data sets?

2. I felt that Section 5.2 on visualizing the trajectories of different optimizers did not have a clear take away message.

3. Except for Figure 1, PHATE is not compared against more commonly used reduction algorithms such as tSNE and UMAP. For example, regarding Figure 2: Could we draw similar conclusions about generalization using other techniques?

---

> ### Author Response · Authors · 2020-11-25
> **Author response to Reviewer 4**
>
> We thank the reviewer for his/her valuable comments. It is true that more analysis settings (architectures and data sets) would help support our claims regarding generalization, and we will focus on that in the future. To our knowledge however, the generalization properties of flat vs. sharp minima are still largely unknown. While it is true that some recent papers have studied this, they are very few in number and the conclusions drawn were still relatively limited.
>
> As for the third comment, we would like to point out that we discuss this in detail in Section 4. More precisely, Figure 1 does contain a comparison of the jump + retrain experiment results embedded using PHATE, PCA, tSNE and UMAP (see Fig. 1 A4, B1, B2, B3). It is similar to what was plotted in Figure 2 but just for a “good generalization” optimum while Fig.2 contains other optima as well. It is clear from those plots that PCA, tSNE and UMAP do not allow the reader to draw similar conclusions about generalization. PCA only captures variance from two directions and makes all trajectories look linear while tSNE and UMAP do not properly preserve inter-trajectory variance, points sampled from the same trajectory are clustered together disregarding the other trajectories and the global structure of the data.

---

### Official Review · AnonReviewer2 · 2020-10-28
**A promising dimensionality reduction technique is applied to DNN trajectories in the weight space, however, more experimental validation is needed to draw meaningful loss landscape implications**

**Rating:** 5
**Confidence:** 4

**Review:**

### 1. Brief summary:
The authors use a new dimensionality reduction technique called PHATE (that was first introduced in a different paper) to study the weight-space positions and training trajectories of several DNN architectures (ResNet, WideResNet) on vision classification tasks (CIFAR-10, CIFAR-100). They use different optimizers (SGD, SGD+Momentum, Adam), study very good and decent (they call them bad) optima, and the solutions to memorization of random labels. They perform two kinds of experiments: 1) perturbing a single optimum and retraining from there, visualizing those trajectories, and 2) visualizing several random init -> optimum trajectories from different inits. They then draw some conclusions from the PHATE projections for the two kind of experiments and different optimizers and kinds of optima.

### 2. Strengths
* I like the introduction of the PHATE algorithm and the authors explanation of its advantages and why it might be a good fit for the weight space trajectory visualization of DNNs.
* The paper is well motivated. Understanding the loss landscape of DNNs is likely very important and still very underexplored.
* I like the comparison to other algorithms (PCA, t-SNE) and the synthetic tree-like structure they visualize in addition to real DNNs to demonstrate the advantageous properties of PHATE.
* I like that the others study the stability of their visualization to retraining and seed change -- that is very good and should be the norm in all papers that make claims that depend on stochasticity.

### 3. Weaker points
1. The scope of the paper is, in my opinion, a bit too limited. A lot of space is used to introduce the PHATE algorithm that is, as I understand it, /not/ a novel contribution of this work.

2. While the visualizations look compelling, the conclusions that are drawn from them are often too strong. I don't see how the visualization can be connected to flatness, dimensionality etc directly. They do serve as a good visual guide, but I feel the paper doesn't establish the connection to those quantities very well. I see that let's say for the random labels and real labels, the projections look different, but how do I link this to e.g. the size of the low-loss basin around those two?

3. The authors flip between a language suggesting that there are some specific dimensions that are being visualized (e.g. "potentially flying off to an orthogonal subspace" and appreciating that the PHATE technique doesn't preserve dimensions in any meaningful way. I would advise to soften the claims to reflect the non-linear, adaptive nature of PHATE and the difficulty of connecting the embeddings to any particular directions.

4. The experiment type 1 = going in random directions off an optimum and retraining back doesn't seem to push far enough. What would happen if I perturbed more? Would the solutions glide back as they do? How does this compare between the real labels and random labels? I'd say this would be a nice validation at least -- if you go to far, you should not go back. If you do, the PHATE projections do not project the way you believe. If you don't, that's interesting on its own and can be used for comparison between cases (Adam, SGD, random labels ...).

5. The Figure 3 results for a single trajectory look uninformative -- what would happen if you did this multiple times from different seeds? Are the Adam paths smoother generically, or just this particular random seed? You should either make the experiment statistically meaningful in some way, or explain what the reader is supposed to take away from it. As is, it doesn't do much.

6. The experiments of type 2: random inits -> training -> optima, look promising. However, they really break my intuition for what PHATE does and if it is at all relevant. My main worry is this: why do the inits end up coinciding? From [1] and [4] it is quite certain that a) the inits are mutually orthogonal to a high degree and b) so are their optimized endpoints. How come your inits get mapped to the same point sometimes, but other times they do not? This really puts your results into question for me, especially because you use the visualizations to make pretty strong claims about the loss landscape structure.

7. (correct me if I am wrong) Only WideResNet and ResNet on CIFAR-10/100 were used in this paper. Those are two pretty similar architectures and two very similar vision tasks. I would appreciate a much broader set of experiments to make sure the claims here hold. For example,
a. what do skip connections do? both architectures here have them, and
b. How would MNIST, FASHION MNIST, SVHN do? Those are cheap to train and easy to use datasets and I would expect the authors do use them
c. How would a simple feed-forward CNN do?
d. What about fully-connected nets? Would they behave the same way?
Empathically, I am /not/ asking for ImageNet -- I know that it is very hard to run and while it would be nice to have, there is no need to use it in every paper. But I would want to see a much broader set of experiments  on other CNN-based architectures as a well as other, potentially weaker datasets. As is, the paper doesn't have the sufficient experimental support to see the generality of the interpretation claims.

### 4. Relevant papers that might be worth adding/exploring
There are some papers that I think might be relevant here and that the authors might want to wish to add / read / explore.

[1] /Large Scale Structure of Neural Network Loss Landscapes/ by Stanislav Fort, Stanislaw Jastrzebski (https://arxiv.org/abs/1906.04724) at NeurIPS 2019 build a geometric model of the low-loss manifolds of DNNs incorporating the observations of 1. connectivity of init to optimum, the high-dim nature of the manifolds and their connectedness. They also visualize the loss landscape on sections that might be relevant here. They also show that they can find N-dimensional surfaces connected N+1 independent optima together, going beyond the 2 optima on a 1-d path.

[2] You might be missing the second paper that established the connectivity between different modes: Garipov, T., Izmailov, P., Podoprikhin, D., Vetrov, D. P., and Wilson,  A. G.  /Loss Surfaces, Mode Connectivity, and Fast Ensembling of DNNs/ at http://arxiv.org/abs/1802.10026

[3] /Measuring the Intrinsic Dimension of Objective Landscapes/ by Chunyuan Li, Heerad Farkhoor, Rosanne Liu, Jason Yosinski (https://arxiv.org/abs/1804.08838) establishes that the loss landscape low-loss manifolds have a low /intrinsic dimension/ that can be interpreted as high d of the manifolds (done in https://arxiv.org/abs/1906.04724)

[4] /Deep Ensembles: A Loss Landscape Perspective/ by Stanislav Fort, Huiyi Hu, Balaji Lakshminarayanan (https://arxiv.org/abs/1912.02757) look at the cosine similarity of the weight space positions of multiple runs as well as a single run. They also visualize the loss landscape between optima on different affine and non-affine sections.

In your justification for why PHASE might be good at visualizing the trajectories you mention that they are low dimensional. There is a number of works showing that it might or might not be the case:
[5] Gradient Descent Happens in a Tiny Subspace by Guy Gur-Ari, Daniel A. Roberts, Ethan Dyer (https://arxiv.org/abs/1812.04754) shows the gradients in a mostly low-D subspace, but the actual learning happens orthogonal to that.
[6] Emergent properties of the local geometry of neural loss landscapes by Stanislav Fort, Surya Ganguli (https://arxiv.org/abs/1910.05929) dissects the Hessian structure and finds low-D signal + high-D noise, where the signal is not in the learning directions but rather forms constraints.
[7] Traces of Class/Cross-Class Structure Pervade Deep Learning Spectra by V. Papyan (arXiv:2008.11865) has a on overview of the low-D structures in the DNN Hessians that might be relevant.


### 5. Summary
I like that you introduced a new dimensionality reduction technique to visualizing loss landscape trajectories and positions for DNNs. The reasons why it might be better than others are compelling. However, I think that the scope of the experiments you provide is limited, the claims you make a bit stronger than would be justified by the results shown, and there are some worrisome features of some of the embedded trajectories that make my question the validity of your overall conclusions. I think the paper has a promise, but it needs a bit more work. **I am open to revising my score** if you address my questions well.

---

> ### Author Response · Authors · 2020-11-25
> **Author response to Reviewer 2**
>
> We really appreciate the fact that the reviewer has taken the time to write a high quality review. We thank him/her for the detailed and extensive comments, helpful insights and very valuable paper recommendations. We feel the reviewer has well identified the strengths of our work while also correctly pointing out its weaknesses.
>
> We have done our best to address some of the questions raised (see responses below) but we recognize that most of the comments made are of a more general nature and addressing those would require a significant rework of the paper. Such comments are greatly appreciated and will be carefully taken into consideration for improving this project into the future but have not been extensively answered here.
>
> Response to “6”: The reason the initial points end up close together in all the plots (but especially in the ones corresponding to the “good generalization” optima) is because of the use of cosine distance. The jump and retrain experiment results consist of 20 re-training trajectories (4 different step-sizes in each of 5 distinct directions from the optimum). Cosine distance is intrinsically based on the angle between two vectors. In the good optimum case, the “flower” shape has 5 “petals” each one corresponding to a different jump direction and all 4 initializations (different step-sizes) are close together. The trajectories converging in a rather monotonous manner towards the middle of the plot (the optimum) strongly suggests that the region surrounding the minimum is a valley.
>
> Response to “7”: In future iterations of this project we will consider more analysis settings (architectures and data sets) which will hopefully help strengthen our conclusions.

---

### Official Review · AnonReviewer1 · 2020-11-05
**Interesting visualization but the interpretation is not quite clear**

**Rating:** 5
**Confidence:** 4

**Review:**


The authors propose to visualize training trajectories with the dimension reduction and visualization tool - PHATE. The authors try to reconstruct manifolds on which the training trajectories lie. It argues that the method captures data characteristics from all dimensions and shows consistent geometric patterns for loss landscape surrounding good and bad generalization optima.


Strength
- The approach embeds multiple training trajectories in one visualization by adopting dimension reduction approaches, which is not explored before. The comparison of multiple dimension reduction approaches is also lacking in previous literatures.
- The paper has a nice review and summarization of previous approaches on loss surface visualization.

Weakness

My major concern is that the insights observed from the methods are not quite straightforward for interpretation and some conclusions are not clear or novel.

- In page 4, the authors claimed “We note that methods that pick random directions...are not able to visualize entire trajectories as the spaces and planes change” this observation is also made in previous work[1] and PCA is used for visualizing the training trajectories. However, this is not necessarily a disadvantage for visualizing the flatness of the minima as discussed follows.

- To characterize the region of loss landscape, the authors propose to adopt “jump and retrain” (setting 1) and this involves visualizing multiple trajectories together. However, the rationale to do this is not quite clear and the interpretation of the visualization is hard. It is not quite clear about the advantage of proposed visualization (Figure 2) on explaining the “flatness” in comparison with previous 1D or 2D visualization with randomized directions. On the other hand, the computation cost (multiple initialization, multiple steps and retraining) for generating those figures could be significant.

- The proposed visualization should be valuable for comparing different trajectories produced by different optimizers as all trajectories can be plotted together, however, the visualization (figure 4) does not convey a clear message about which optimizer (trajectory) is better in performance (generalization).

- The individual trajectory visualization (Figure 3) in sec 5.2 seems not to differ too much with the PCA based visualization [1][2]. What is the takeaway message from Figure 3? It is not clear how the observation “the resulting parameters appear to be further away from the initialization” is made from the figure.

Questions

In section 5.3, the authors claim there are consistent and distinct patterns for “good” and “bad” minima. Is there a smooth transition between the “good” and “bad”? Will the pattern differ with a much better or worse initialization? e.g., what is the pattern for random initialization (which is the “worst”)?


[1] Li et al, Visualizing the Loss Landscape of Neural Nets, NIPS 2018.
[2] Eliana Lorch. Visualizing deep network training trajectories with pca. ICML Workshop on Visualization for Deep Learning, 2016.

---

> ### Author Response · Authors · 2020-11-25
> **Author response to Reviewer 1**
>
> We thank the reviewer for the constructive criticism and the insights provided. Below is a list of responses to some of the comments made.
>
>
> Comment: “[...] PCA is used for visualizing the training trajectories. However, this is not necessarily a disadvantage for visualizing the flatness of the minima as discussed follows.”
>
> Response: Getting good visualizations utilizing PCA requires carefully choosing linear subspaces of the parameter space, yet the resulting trajectories disregard variations in all other dimensions than the chosen 2-3 directions. Furthermore, the PCA visualization method mentioned was only used to visualize one individual trajectory at the time and the linear projection subspace was chosen accordingly. If one wants to plot multiple trajectories with a PCA method, it is likely that only the main direction of variance is identified for each trajectory. The resulting plot would look somewhat similar to our Figure 1 (A4) where inter-trajectory variance was somewhat captured but intra-trajectory variance was almost completely ignored.
>
>
> Comment: “However, the rationale to do this (the “jump and retrain experiment”) is not quite clear and the interpretation of the visualization is hard. It is not quite clear about the advantage of proposed visualization (Figure 2) on explaining the “flatness” in comparison with previous 1D or 2D visualization with randomized directions. On the other hand, the computation cost (multiple initialization, multiple steps and retraining) for generating those figures could be significant.”
>
> Response: We agree with the reviewer that our method has a higher computational cost. However, given how little we know about the loss landscape of ANNs, at this point we believe using such methods is necessary for better understanding the loss landscape.  The past linear 1D or 2D visualization methods may generate misleading results.  For example,  the non-convexities seen in the one or two randomly chosen directions only exist in the specific  1D or 2D subspace and may never be visited during navigation in the high-dimensional space. Such methods do not tell us what is going on in the remaining millions of directions. Similarly, while a region might appear “flat” in a specific 2D visualization nothing guarantees that if two other random directions were chosen the region wouldn’t appear “sharp”. The main point of our work is that it is insufficient to look at a low-dimensional subspace of the parameter space, and we try to explore the parameter space in a more extensive manner using a non-linear method. Our method allows for capturing variance from a multitude of directions and ensures we do not miss significant geometric characteristics from other dimensions which would have not been considered by linear 1D or 2D methods.
>
>
> Comment: “The proposed visualization should be valuable for comparing different trajectories produced by different optimizers as all trajectories can be plotted together, however, the visualization (figure 4) does not convey a clear message about which optimizer (trajectory) is better in performance (generalization).”
>
> Response: Section 5.2 was not aimed at showing which trajectory is better in performance but rather at qualitatively comparing the trajectories and their characteristics using PHATE.

---

### Decision · Program_Chairs · 2021-01-07
**Final Decision**

**Decision:**

Reject

**Comment:**

The reviewers and I agree that the paper is well motivated and that there are good comparisons to prior work. However, the scope of the paper is rather limited, and there were some doubts about the overall conclusions and whether the current results fully support them. As such, I cannot recommend the paper for publication.